



# A wedge strategy for mitigation of urban warming in future climate scenarios

Lei Zhao[1,2,3], Xuhui Lee[2,3], Natalie Schultz[3]

[1]Program in Science, Technology and Environmental Policy (STEP), Woodrow Wilson School of Public and International Affairs, Princeton University, Princeton, New Jersey, 08540, USA
[2]Yale-NUIST Center on Atmospheric Environment, Nanjing University of Information Science and Technology, Nanjing, 210044, China
[3]School of Forestry and Environmental Studies, Yale University, New Haven, Connecticut, 06511, USA

*Correspondence to*: Lei Zhao (lei.zhao@princeton.edu)

**Abstract.** Heat stress is one of the most severe climate threats to the human society in a future warmer world. The situation is further compounded in urban areas by the urban heat island (UHI). Because the majority of world's population is projected to live in cities, there is a pressing need to find effective solutions for the high temperature problem. We use a climate model to investigate the effectiveness of urban heat mitigation strategies: cool roofs, street vegetation, green roofs, and reflective pavement. Our results show that by adopting highly-reflective roofs, almost all the cities in the United States and southern Canada are transformed into "white oases" at midday, with an average oasis effect of -3.4 K in the summer for the period 2071-2100, which offsets approximately 80% of the greenhouse gas (GHG) warming projected for the same period under the RCP4.5 scenario. A UHI mitigation wedge consisting of cool roofs, street vegetation and reflective pavement has the potential to eliminate the daytime UHI plus the GHG warming.

## 1 Introduction

Heat stress associated with climate change is projected to cause a substantial increase in human mortality (Patz et al., 2005; Huang et al., 2011) and a large reduction in workplace productivity (Dunne et al., 2013; Zander et al., 2015). These risks are further amplified for urban populations because of the urban heat island effect (Grimmond, 2007). Because more than 50% of the world's population currently lives in cities, and that number is projected to increase to 70% by year 2050 (Heilig, 2012), there is a pressing need to find effective solutions for urban heat stress. In recent years, urban climate agendas have broadened beyond carbon management, which brings marginal heat relief to urban residents (Revi et al., 2014), to urban climate adaptation. It is now recognized that in addition to the traditional emphasis on preparedness to cope with heat stress (Stone et al., 2012), urban adaptation should include active modifications to the urban landscape to reduce urban temperatures (Rizwan et al., 2008; Castleton et al., 2010; Santamouris, 2013; Bowler et al., 2010; Jacobson, 2009).





Field trials have demonstrated the cooling benefits of street vegetation (Shashua-Bar and Hoffman, 2000; Mackey et al., 2012), bright pavement (Rosenfeld et al., 1995; Mackey et al., 2012), green roofs (Jim and Peng, 2012), solar roofs (Dominguez et al., 2011), and cool roofs (Ismail et al., 2011) at the building scale. Street vegetation and green roofs provide localized cooling through enhanced evaporation, and for street vegetation by increasing surface roughness and therefore

convection efficiency. Bright pavement and cool roofs, both characterized by a higher albedo than traditional materials, cool the urban environment by reflecting a higher portion of solar radiation away from the surface. Solar roofs produce localized cooling through their conversion of solar energy to electricity.

Although it is financially impossible to conduct field experiments on these methods at the city scale, modeling studies have

evaluated the potential of citywide implementation of these methods in reducing urban ambient temperatures. A majority of the modeling studies deploy a mesoscale weather forecast model, such as the Weather Research Forecasting (WRF) model (Skamarock et al., 2008), and model simulations are performed for a business-as-usual case and for one or more mitigation scenarios, including tree planting (Stone et al., 2014; Stone et al., 2013), green roof (Li et al., 2014; Stone et al., 2014), cool roof (Georgescu et al., 2014), and rooftop solar panels (Salamanca et al., 2016).

The computational demand of mesoscale modeling limits long-term and large-scale simulations, with most of these studies restricting their analysis to high-impact heatwaves in a single city (Rosenzweig et al., 2009; Li et al., 2014). Recently, studies using a mesoscale modeling approach have begun to scale up the geographic scope of analysis, comparing and evaluating urban heat mitigation methods across multiple (up to 3) cities in contrasting regional climates (Stone et al., 2014;

Vargo et al., 2016) and even over the continental United States (Georgescu et al. (2014).

A second modeling strategy, using subgrid output from a global climate model, overcomes the computational constraints of mesoscale models, allowing for the continental or global-scale analysis of UHI and heat mitigation strategies over climatological timescales (Oleson et al., 2010c; Hu et al., 2016; Jacobson and Ten Hoeve, 2012). Although climate models

cannot resolve local-scale processes, such as advection between adjacent urban and rural land in the same grid cell, they can capture the urban effects on the large-scale dynamics of the atmosphere. Additionally, GCMs can simulate the interactions of GHG-induced warming and the biophysical drivers of urban heat islands and heat mitigation methods. As urban heat mitigation is becoming an integral part of the climate adaptation agenda, information from GCMs can help inform city planners on the effectiveness of various mitigation strategies across a range of climate conditions and under different climate

scenarios.

In this study, we use an urban climate model to assess the effectiveness of urban climate mitigation methods in future warmer climates. Our specific objectives are to: (1) investigate the urban heat island intensity under current and future climate scenarios for three climate regions in the U. S. and southern Canada, (2) quantify the effectiveness of several



mitigation strategies (street vegetation, green roof, cool roof and bright pavement) in offsetting urban warming, and (3) estimate aggregated temperature reduction potential of these strategies using an UHI mitigation wedge approach. Our research complements the studies by Georgescu et al. (2014), Rosenzweig et al. (2009), and Stone et al. (2014) in that we are also interested in quantifying the effectiveness of multiple UHI mitigation strategies, both individually and collectively,

across a diverse range of climate conditions. Here, we expand the analysis to several climate scenarios and a large number of cities (57 cities) to understand the interactions between the UHI biophysical drivers, GHG-induced warming, and influence of local background climate. We propose a new method to assess the mitigation strategies, which is based on a theoretical understanding of the surface energy balance and is unconstrained by computational demand.

## 2 Materials and Methods

We used a global climate model to simulate the UHI and to quantify the cooling potential of urban heat mitigation strategies: cool roofs, street vegetation, green roofs, and reflective pavement. As will be discussed in more detail in the following subsections, the simplified representation of urban areas in the model does not directly allow for online assessment of each of the mitigation strategies. Therefore, in addition to the online simulations to assess the mitigation potential of cool roofs, we used an offline attribution method to quantify the mitigation potential of green roofs and reflective pavement, and a two-

member interpolation method to quantify the cooling potential of street vegetation (Table 1) for three climate scenarios. The offline calculations are based on the diagnostic variables of the surface energy balance produced by the model.

### 2.1 Climate model and simulations

We used the Community Earth System Model (CESM) (Hurrell et al., 2013) to simulate the UHI for cities in the United States and southern Canada. The land surface processes are represented by the Community Land Model (CLM, version 4.0),

the land component of CESM (Oleson et al., 2010b). In CLM 4.0, the land surface heterogeneity is represented as a nested hierarchy of subgrid levels. The model grid cell consists of up to five land units: vegetated land, glacier, wetland, urban, and lake, all of which are driven by the same atmospheric forcing. In the CESM architecture, fluxes and state variables are computed at each land unit level, and then area-weighted to grid cell means before being passed to the atmosphere model. We use data from the urban and vegetated land units in this analysis, to represent urban and rural land cover types,

respectively. The urban land unit is represented in a canyon conceptual structure that consists of roof, sunlit wall, shaded wall, and pervious (bare soil) and impervious (road, sidewalk and parking lot) ground (Oleson et al., 2010a). The vegetated land unit is comprised of up to 15 plant functional types and bare soil.

The CLM was run under three climate scenarios: current climate, Representative Concentration Pathway (RCP) 4.5, and

RCP8.5. These three simulations comprise the control (CTR) group of simulations, where all the urban parameters were kept as the default values prescribed by the model. For the current climate, the model was driven by a reconstructed climatology





from 1972 to 2004 (Qian et al., 2006). For the two future scenarios, the model was forced by atmospheric outputs from fully-coupled runs of the CESM (years 2005 to 2100). This model setup is a shortcut to the fully coupled mode and can be considered as a simplified retrieval of the surface climate variables from the fully coupled runs. The results should be nearly identical to those obtained from the coupled simulations because the impacts of large-scale feedbacks represented in a coupled CESM, such as large-scale dynamics and ocean-air feedbacks, are preserved by the atmospheric variables.

This model configuration cannot simulate the dynamic impact of changes to the urban land on the atmosphere. However, because the urban land unit in CLM comprises only a small areal fraction of each grid cell (< 0.4%), changes in urban temperature would lead to negligible changes in grid cell mean temperature. Therefore, the dynamic feedbacks between changes in urban land and the atmosphere are negligible. Studies that fully couple CLM to the atmosphere conclude that modification of the urban albedo has negligible effects on the regional and global climate (Hu et al., 2016; Zhang et al., 2016).

We used thirty-three years (1972 - 2004) of data in the current climate and the last thirty years (2071 - 2100) of data in the two future scenarios to compute climatological mean temperatures and the UHI intensity. We analyzed the UHI intensity for 57 selected cities, representing three Koppen-Geiger climate zones: temperate climate (24 cities, eastern and southern U. S.), continental climate (20 cities; northern U. S. and southern Canada), and dry climate (13 cities; arid and semiarid western U. S.). Model outputs at 13:00 and 01:00 local time were used to represent daytime and nighttime conditions, respectively. Our analysis was restricted to summer (June-August) results.

In this analysis, we use the surface temperature instead of the screen-height air temperature in our assessment of UHI and the temperature mitigation potential of cooling strategies. In other words, the UHI intensity ($\Delta T$) is defined as the surface UHI, the difference in radiative surface temperature between urban and nonurban subgrid land units in the grid cell where the city is located. This is different from the air UHI which is defined using screen-height air temperature. These two UHI definitions differ in a variety of aspects. First, the air UHI can be affected by the local air and landscape conditions and thus is susceptible to the inhomogeneous urban landscape. The surface UHI, instead, is a spatial average. Although the surface temperature does not necessarily match the human experience of warmth while walking across the rural-urban boundary, it is more stable as a metric of the city-scale microclimate and can be directly validated against satellite observations over large areas, especially when compared across different cities over diverse local climate conditions. Second, the air UHI is more tightly related to heat stress and heat exposure assessment (Oleson et al., 2015) than surface UHI. However, the surface UHI has a firm theoretical basis that underpins the wedge method for assessment of multiple mitigation strategies. It has been shown from the surface energy balance principle that the overall surface UHI can be estimated by linear supposition of different biophysical contributions (Zhao et al., 2014). The linear supposition property, that the overall temperature reduction due to simultaneous implementation of several mitigation methods, can be approximated by the sum of cooling benefits from





implementation of these methods one at a time, is the underlying basis for the UHI mitigation wedge approach used in this study. In addition, this linear supposition seems to be a robust property for the air UHI as well. We will present evidence from published UHI mitigation studies using the air UHI (Taha et al., 1997; Georgescu et al., 2014; Stone et al., 2014; Salamanca et al., 2016).

## 2.2 Offline UHI attribution

Defining UHI by surface temperature allows us to estimate the contribution of different biophysical mechanisms to the overall surface UHI from surface energy balance principles (Zhao et al., 2014). In addition to $\Delta T$, the model also produces diagnostic variables of the surface energy balance. In this analysis, these diagnostic data served two purposes. First, they

were used to attribute the UHI ($\Delta T$) to contributions from changes in surface biophysical parameters including albedo, evaporation, convection efficiency, heat storage and anthropogenic heat addition (Zhao et al., 2014). This offline attribution analysis serves to examine the linear supposition of different biophysical contributions which is the basis underlying the UHI mitigation wedge approach used in this study. Second, they allowed us to estimate the cooling benefit of two of the four UHI mitigation methods (green roof, and reflective pavement; details described below).

### 2.3 Temperature mitigation strategies

### 2.3.1 Cool roofs

To investigate the mitigation potential of cool roofs, an additional set of simulations was conducted for each of the three climate scenarios. In this group of simulations (WHT), the roof albedo was increased to 0.88, the value of the U.S. EPA Energy Star SOLAREFLECT coating material after three years of wear and aging (Georgescu et al., 2014). This value is

slightly lower than the initial albedo of 0.89, indicating the material's high capability of maintaining high solar reflectance (https://downloads.energystar.gov/bi/qplist/roofs_prod_list.pdf?8ddd-02cf). The cooling benefit of the cool roofs was determined by comparing the urban surface temperature in the WHT simulation with that in the CTR simulation, where the roof albedo was kept as the default value (0.18 – 0.37), under each climate scenario.

Of the 57 selected cities, the average roof areal fraction is 48.8%, the average default roof albedo is 0.29, and the average citywide albedo is 0.18. After the deployment of white roofs, the average citywide albedo is increased to 0.47.

To help the reader visualize the changes brought by the cool roofs to the urban landscape, we created two animations using a 3D point cloud and image mosaic dataset acquired by an unmanned aerial vehicle over an urban neighborhood in

Switzerland (www.sensefly.com). In Animation Movie S1, the roof pixels retain their natural reflectance values. In





Animation Movie S2, these pixels were replaced with saturation reflectance values to simulate the SOLAREFLECT coating material.

### 2.3.2 Street vegetation

It is not possible to directly evaluate the cooling benefit of street vegetation with CLM because vegetation is not explicitly represented in the urban land unit in the current version of CLM. Here, we used a simple two end-member interpolation method to calculate the surface temperature change δT associated with street vegetation, as

$$\delta T = -V \times \Delta T_C \qquad (1)$$

where $V$ is the areal fraction of street vegetation in the urban land unit, and $\Delta T_C$ is the UHI intensity from the CTR run. A negative δT indicates cooling effect, and vice versa. This simple linear method satisfies the two end members: at 0% vegetation, there is no temperature reduction, and the UHI intensity is the original ΔT; at 100% vegetation, the urban land would be completely converted to the rural landscape, and thus the cooling benefit should totally offset the original ΔT. The calculation assumes that street vegetation consists of native plant species having the same species compositions in the adjacent rural land. For each of the selected cities, $V$ is the areal fraction of pervious surface in the urban land unit prescribed by the model. The average $V$ value for these cities is 30%.

This method yields the net cooling of all the biophysical effects associated with street vegetation, including changes in albedo, convection efficiency, evaporation, and storage. It should be noted here that this method may slightly overestimate the vegetation cooling benefit for two reasons. First, the UHIs from the CTR simulations also include anthropogenic heat contribution. According to the offline attribution results, the anthropogenic heat contribution is about 20% (Figure 1). Second, street vegetation is inherently not identical to their rural counterparts due to the radiative trapping by urban canyons via the "canyon effect" (Wang, 2014; Ryu et al., 2016). Therefore, same amount of vegetation inside urban canyons may not induce as much cooling as it would in the adjacent rural land.

### 2.3.3 Green roofs

We calculated the cooling benefit of green roofs using the diagnostic surface energy balance data, as

$$\delta T = \frac{\lambda_0}{1+f}(\Delta a)K_\downarrow + \frac{-\lambda_0}{(1+f)^2}(R_n^* - Q_s + Q_{AH})(\Delta f_2) \qquad (2)$$

with,

$$R_n^* = (1-a)K_\downarrow + L_\downarrow - (1-\varepsilon)L_\downarrow - \varepsilon\sigma T_a^4 \qquad (3)$$

$$f = \frac{\lambda_0 \rho c_p}{r_a}\left(1 + \frac{1}{\beta}\right) \qquad (4)$$

$$\Delta f_2 = \frac{-\lambda_0 \rho c_p}{r_a}\left(\frac{\Delta\beta}{\beta^2}\right) \qquad (5)$$





where $T$ – surface temperature, $\lambda_0$ – local climate sensitivity ($= 1/4\varepsilon\sigma T^3$), $f$ – energy redistribution factor, $R_n^*$ – apparent net radiation, $\rho$ – air density, $C_p$ – specific heat of air at constant pressure, $r_a$ – aerodynamic resistance to heat diffusion, $\beta$ – Bowen ratio, $a$ – surface albedo, $K_\downarrow$ – incoming solar radiation, $L_\downarrow$ – incoming longwave radiation, $\varepsilon$ – surface emissivity, $\sigma$ – Stefan-Boltzmann constant, $T_a$ – air temperature at the blending height. Equation (2) is obtained by differentiating the
surface temperature solution given in Lee et al. (2011).

The Bowen ratio of green roofs was the average Bowen ratio value for grassland calculated at the subgrid level by the CLM for the grass plant functional type with its own soil column (Schultz et al., 2016). This implies that the water-conserving native grass is used for green roofs. The citywide Bowen ratio change was the area-weighted average of the green roof
Bowen ratio and the Bowen ratio of other surface components in the urban land unit. The Bowen ratio change ($\Delta\beta$) was the difference in Bowen ratio between urban land units with and without green roof installation.

The green roofs were assigned the average warm-season grassland albedo value of 0.20 (Bonan, 2008). The citywide albedo was the area-weighted average of the green roof albedo and the albedo of other surface components in the urban land unit.
The albedo change ($\Delta a$) was the difference between the city with and without green roof installation. The average $\Delta a$ of the 57 cities was 0.01.

The change in surface roughness induced by green roofs is minimal, and thus was omitted in this analysis.

### 2.3.4 Reflective pavement

The same offline method was used to calculate the effect of reflective pavement, as

$$\delta T = -\frac{\lambda_0}{1+f}(\Delta a)K_\downarrow \qquad (6)$$

where $\Delta a$ is citywide albedo change associated with the use of reflective pavement. We increased the pavement albedo from the default value of 0.13 to 0.25 as recommended by Akbari et al. (2012). The areal fraction of pavement was on average 20.2%. The average citywide albedo of the 57 cities was increased by 0.04.

## 3 Results and Discussion

### 3.1 Offline UHI diagnostics

An example of the attribution result is given in Figure 1 for the period 2071-2100 under RCP4.5. The offline ΔT (the sum of the component contributions) shows good agreement with the online ΔT (the surface temperature difference between the urban and the vegetation land unit computed directly by the model). The offline ΔT is slightly lower than the online ΔT
because high-order terms are ignored in the linearization of the surface long-wave radiation term of the energy balance





equation and nonlinear interactions among the component contributions are omitted (Zhao et al., 2014). Good agreement was also obtained for the current climate conditions (Zhao et al., 2014), implying that nonlinear interactions among the various UHI contributors are small.

This linear supposition property of different biophysical contributions is the theoretical basis underpinning the wedge approach that we use to quantify the overall effectiveness of multiple UHI mitigation strategies. Different mitigation strategies affect the urban surface temperature through these biophysical contributions. Combining different UHI abatement actions is actually adding up the perturbed biophysical contributions in the process level. Linear supposition of various biophysical contributions states that different mitigation strategies add up linearly as well. Therefore, good agreement

between online ΔT and offline ΔT (Figure 1) supports the idea that different UHI mitigation strategies, as long as their deployments are not mutually exclusive, act nearly linearly when quantified collectively.

The validity of our offline method is further supported by the excellent agreement between the online and offline ΔT for a total of 18 combinations of climate zone, roof choice and climate scenario (Figure 2). In the offline calculation, changes in

the citywide albedo from CTR to WHT were computed from the modeled reflected solar radiation and the incident solar radiation in the urban land unit. The citywide albedo changes were then substituted into Equation 6 (the offline method for cool roof is equivalent to the method for reflective pavement). Other component contribution terms were kept unchanged in the offline UHI attribution equation, because cool roofs only alter the radiation contribution to ΔT in the surface energy balance. The $R^2$ value of the linear correlation between the two sets of ΔT calculations is 0.99 and the mean bias (offline ΔT

minus online ΔT) is 0.1 K.

### 3.2 Future UHI under RCP scenarios

We find strong urban climate change signals under both RCP scenarios. Near the end of this century, the average summer daytime ΔT is projected to increase by 0.9 ± 0.2 K (mean ± 1 standard error) and the nighttime ΔT by 1.5 ± 0.2 K under the RCP4.5 scenario (Figure 3a & b). This is in addition to the GHG-induced summer surface temperature increase of 4.2 K and

2.6 K from the current temperature of 32.7°C and 18.0°C in the daytime and nighttime, respectively, for years 2071-2100. The increase in ΔT is partly the result of higher anthropogenic heat release, primarily from air-conditioning (AC) energy use to cope with the GHG-induced warming (Table 2; (Hu et al., 2016; Oleson et al., 2011). The contribution of anthropogenic heat to the UHI intensity in the current climate is 0.4 K and 0.02 K in the daytime and nighttime, respectively, and is increased to 0.8 K and 0.7 K, respectively in the RCP4.5 scenario. This conclusion is supported by the offline UHI

attribution analysis described by Zhao et al. (2014). The larger increase in the nighttime ΔT than the daytime ΔT confirms the well-established fact that anthropogenic heat is a dominant contributor to urban warming at night (Oke, 1982).




In the current climate, the spatial variations of daytime ΔT across the climate zones are controlled by the precipitation regime, being lowest in the dry climate and highest in the humid temperate climate (Zhao et al., 2014). This zonation is preserved in the future warmer world (Figure 3a).

### 3.3 Effectiveness of the mitigation strategies

**3.3.1 Cool roof**

Adoption of the bright, reflective SOLAREFLECT roof material transforms almost all the cities into cold islands in both the current and the future climates, as indicated by the negative daytime ΔT (Figure 3c, Figure 4c). In other words, these cities become isolated "white oases" surrounded by a hot landscape. Under the RCP4.5 scenario, the average daytime surface temperature reduction in the WHT simulation is $6.5 \pm 0.3$ K for the period 2071-2100, in reference to the CTR simulation for

the same period (Supplementary Table S1). There is a discernible spatial pattern in the oasis effect, which follows the climatological wetness gradient across the continent (Figure 4c). The strongest oases are located in the dry region where the urban land is $6.2 \pm 0.4$ K cooler than the surrounding rural land (Figure 3c, Figure 4c). For comparison, the average oasis effect for the cities in the humid temperate region is -2.6 ± 0.5 K. The average ΔT of all the cities is -3.4 ± 0.3 K, enough to offset 80% of the GHG warming. The cooling benefit of reflective roofs is similar under the RCP8.5 scenario

(Supplementary Table S1, Figure 3c).

Our results indicate a stronger cooling effect of reflective roofs than those estimated by previous research. Studies using the WRF model showed a cooling effect of 0.2 – 2.0 K in screen-height air temperature with reflective roof materials of different albedos (Georgescu et al., 2014; Li et al., 2014; Stone et al., 2014). A case study of the California South Coast Air Basin area

using a mesoscale model coupled with an urban airshed model showed that a citywide albedo increase by 0.30 can reduce the air temperature by up to 4.5 K in the mid-afternoon (Taha et al., 1997). However, Rosenzweig et al. (2009) reported very modest cooling effects (0.3 – 0.6 K) from cool roofs with albedo of 0.5 in New York City using a regional climate model. Studies using global climate models also report local cooling effects, ranging from a global average of 0.02 K (Jacobson and Ten Hoeve, 2012) to 0.4 K (Oleson et al., 2010c) decrease in air temperature with different prescribed cool roof albedos.

Apparently, these estimates vary with the choice of cool roof albedo, as well as the roof space available for cool roof installation (Akbari et al., 2012; Rosenzweig et al., 2009).

There are three reasons why we observe a stronger cooling effect compared to previous studies. First, the albedo of the cool roof in our study (0.88) is at the higher end of the range (0.3 – 0.9) used in previous modeling studies. Second, we define ΔT

using radiative surface temperature rather than screen-height air temperature. The radiative surface temperature is more sensitive to the surface radiation balance than the air temperature. Using a mesoscale weather model, Li et al. (2014) showed that, when measured by radiative surface temperature, the cooling effect of cool roofs was 3 – 4 K, much higher than that of





0.5 K measured by 2-m air temperature. These results are consistent with observational studies using radiative surface temperature. Using LANDSAT albedo and radiative temperature observations, Mackey et al. (2012) detected a surface cooling effect of 5.0 K with an albedo increase of 0.16. Gaffin et al. (2012) tested three generic white membranes in New York City and found that the radiative temperature of the white surfaces was on average 23.6 K lower than a black surface

during peak sunlight times. Third, our analysis and results are restricted to midday hours (13:00 local time) and summer months (June-August), rather than the annual mean values as reported in previous studies. The rationale behind this restriction is that heat stress mitigation is a more pressing need in the summer (Rosenzweig et al., 2009). Because surface incoming solar radiation peaks in the summer noon time, the reflected solar radiation by cool roofs would peak as well. The cooling effect induced by cool roof increases as the reflected solar radiation increases. Therefore, the reductions in ΔT

reported here are likely the maximum potential of urban heat mitigation by cool roofs.

Our results are based on the albedo value of roof coating material after three years of use. The long-term weathering and aging problem of the reflective material has not been taken into account. Although the albedo of the SOLARFLECT coating material decreases only slightly from 0.89 to 0.88 after three years, its long-term performance of retaining solar reflectance is

unknown. In addition, none of the current roofing materials can last 96 years (2005 - 2100) while preserving high albedos. Recoating or rerooting seems necessary for long-term use. Therefore, our results should be interpreted as an upper bound of the cooling potential of cool roof strategy.

### 3.3.2 Street vegetation

Street vegetation is less effective than cool roofs in terms of temperature reduction. The daytime surface temperature

reduction is 1.3 ± 0.2 K in the temperate climate and 0.3 ± 0.1 K in the dry climate under the RCP4.5 scenario. The cooling under the RCP8.5 scenario is 1.1 ± 0.1 K in the temperate climate and 0.3 ± 0.1 K in the dry climate (Supplementary Table S1). The cooling effect is weaker in the dry climate compared to that in the temperate climate. One factor that limits the mitigation potential is the area available for vegetation planting. In the model domain, the pervious surface that can be converted to vegetation cover occupies 25% to 45% of the urban land. Our calculation assumes that street vegetation consists

of native species that have adapted to local soil moisture conditions and can grow without additional water supply. Enhanced cooling brought by irrigation is not considered.

As mentioned previously, our method for calculating the cooling potential of street vegetation may slightly overestimate the cooling benefit because of the anthropogenic heat contribution to the UHI and the radiative trapping effects by urban

canyons. Our offline attribution results show that anthropogenic heat contribution is about 20% (Figure 1). Ryu et al. (2016) showed that the trees in the urban canyon decreased the sensible heat flux and increased the latent heat flux by roughly a similar amount. However, integrating all these considerations into our results would not change the point that street vegetation reduces urban surface temperature less effectively in comparison to cool roofs.





The cooling effects of street vegetation estimated in this study are generally consistent with previous studies. Rosenzweig et al. (2009) found that planting trees in open spaces and along streets reduces the mid-afternoon air temperature by 0.2 – 0.6 K in neighborhoods of New York City. Luley and Bond (2002) reported that in their maximum scenario, in which all urban

grass is replaced by trees, an up to 1.0 K reduction in air temperature is expected in Manhattan, New York City on a summer afternoon. By forcing a mesoscale weather forecast model with atmospheric conditions generated by a global climate model, Stone et al. (2014) found that under the "business as usual" climate change scenario, increasing green vegetation in urban public areas reduces the urban daily average air temperature by 0.1 – 0.3 K. Consistent with our results, they also reported a weaker vegetation cooling effect over a city in the dry climate compared to the cities in the temperate climate. In extreme

scenarios, in which the urban center of Atlanta, Georgia is fully substituted by forest, Stone et al. (2013) found a reduction in air temperature of 0.5 – 1.5 K. Generally, the cooling benefit varies with different cities because of the area available for tree planting. Our estimates fall at the higher end of the range reported previously, partly because the temperature difference is measured by surface radiative temperature.

### 3.2.3 Green roof

Green roofs generate modest cooling. The average daytime surface temperature reduction is 1.6 ± 0.2 K for the 57 cities under the RCP4.5 and RCP8.5 scenarios (Supplementary Figures S1c and g; Supplementary Table S1). According to our offline UHI attribution, the albedo effect of green roofs is negligible (Supplementary Table S1). Even though the albedo of green roofs (0.20, the average warm-season grassland albedo; Bonan, 2008) is lower than the average default roof value (0.29), the citywide albedo is decreased only slightly (by 0.01) in comparison to the CTR simulations. This is in sharp

contrast to the WHT simulations where the citywide albedo is increased by an average amount of 0.29. Therefore, the contribution of green roofs to the temperature reduction comes from enhanced evaporation. Once again, we assume in our calculation that water-conserving native grass is planted on the roofs to minimize irrigation demand; this is a fair assumption because water will be a scarce resource in future cities (McDonald et al., 2011). Cities that can afford rooftop irrigation (Georgescu et al., 2014; Bass et al., 2003) are expected to gain more cooling than predicted here.

These results are comparable with the green roof cooling effects reported by previous studies. Building-scale observational studies have shown a relatively large range of temperature reduction. A case study evaluating the "green" and "white" policy in Chicago, Illinois found that green roofs can reduce the building-top temperature by 0.3 – 2.6 K according to the LANDSAT satellite observations (Mackey et al., 2012). Ismail et al. (2011) used an experimental approach on a test building

in Malaysia and showed a 4.6 K reduction in the surface temperature around noontime between a green roof and a black bare roof. Mesoscale modeling studies have also shown cooling estimates ranging from 0.2 – 0.8 K if measured by air temperature (Rosenzweig et al., 2009; Stone et al., 2014) and 3 – 4 K if measured by surface temperature (Li et al., 2014).





All these results, together with our study, confirm that a major factor to determine the mitigation potential of green roof is water availability.

### 3.3.4 "White oasis" effect

Our results favor cool roofs as the preferred method for urban heat mitigation. Although empirical data are not yet available
to validate the white oasis effect, some lessons can be drawn from studies of green oases. Local cooling has been observed for green oases as small as several hectares in area (Kai et al., 1997). A typical green oasis effect ranges from -1 K to -7 K (Potchter et al., 2008), comparable in magnitude to the white oasis effect reported here. The strongest green oasis effect is also found in dry climates (Potchter et al., 2008). In the case of green oases, surface cooling by evaporation typically results in a stable inversion air layer above the ground, which would severely limit air pollution dispersion and worsen air quality.
In contrast, unstable lapse conditions generally prevail over white oases (Supplementary Figure S2). In other words, implementation of cool roofs may decrease the dispersion capacity of urban air due to a reduction in the mixed layer depth (Georgescu et al., 2012), but not nearly to the extent of a stable stratification brought by the green oasis effect.

An explanation to this phenomenon can be achieved using the big-leaf model (Sellers et al., 1996) solution of surface
temperature shown below:

$$T_s - T_a = \frac{r_a + r_c}{\rho_d c_p} \cdot \frac{\gamma(R_n - G) - \rho_d c_p D/(r_a + r_c)}{\Delta + \gamma(r_a + r_c)/r_a} \qquad (7),$$

where $T_s$ – surface temperature, $T_a$ – air temperature at a reference height, $r_a$ – aerodynamic resistance, $r_c$ – surface resistance; $R_n$ – net radiation; $G$ – storage heat; $D$ – water vapor pressure deficit; $\Delta$ – slope of the saturation vapor pressure curve; $\gamma$ – psychrometric constant. Equation (7) shows that the sign of $(T_s - T_a)$ is determined by the sign of the second
numerator on the right-hand side, $-\rho_d c_p D/(r_a + r_c)$. In green oases in dry climates, $D$ is a large number and the surface resistance is small due to irrigation, both factors combined yielding a negative sign of numerator of the term on the right side of the equation. Therefore, an inversion $(T_s < T_a)$ typically prevails in the surface layer over a green oasis. Over white oases, the available energy $(R_n - G)$ is reduced somewhat, but because the surface resistance is very large in urban land, the numerator of the term on the right-hand side of Equation (7) stays positive. Therefore, unstable lapse conditions $(T_s > T_a)$
should be expected over white oases.

### 3.4 Mitigation wedges

On the global (Pacala and Socolow, 2004) and urban (Creutzig et al., 2015) scales, the highest carbon mitigation potential is achieved by using the strategy of mitigation wedges or combining the incremental benefits of a number of abatement actions. Because nonlinear interactions among the various biophysical UHI contributors are small (Zhao et al., 2014; Figure 1), a
similar strategy can be used to estimate the aggregated potential of multiple urban temperature reduction methods. Figure 5 shows the additive benefit of three UHI mitigation wedges: cool roof (or green roof), street vegetation and reflective





pavement. Because cool roof and green roof are mutually exclusive under the assumption of a 100% penetration rate, we did not combine the cooling benefit of cool roof and green roof in a single scenario. The urban daytime surface temperature is on average 2.4 K greater than the rural background temperature in the current climate. Near the end of the century, the urban temperature will be 7.3 K greater under the RCP4.5 scenario than the current background temperature due to the GHG

warming (4.2 K) and the UHIs (3.1 K). The combined wedges with cool roof (Figure 5a) provide greater cooling benefit than the ones with green roof (Figure 5b). The total cooling effect of the three mitigation wedges with cool roof is 8.0 K, essentially eliminating all the UHI effect plus the GHG warming.

The example given in Figure 5 is the most optimistic scenario. Because roof space is a "precious" resource that must

accommodate other competing needs, a 100% conversion to white roofs is probably impractical. However, we can linearly scale the end-member results (Supplementary Table S1) to estimate the cooling benefits of other wedge combinations. For example, a wedge strategy consisting of 50% cool roof, 50% green roof, 100% street vegetation and 100% reflective pavement will bring a total temperature reduction of 5.7 K (cool roof 3.3 K + green roof 0.8 K + street vegetation 0.9 K + reflective pavement 0.7 K).

Conversion of roof space to solar photovoltaics (PV) can also cool the urban air (Hu et al., 2016; Salamanca et al., 2016). A typical solar PV has an albedo of 0.10 and a futuristic electricity conversion efficiency of 25% (Hu et al., 2016), giving a heat removal rate equivalent to having a surface albedo of 0.33, which is 0.04 more than the average default roof albedo value. Scaling the WHT simulation result proportionally by the albedo increase, we estimate that the solar PV will lower the

surface temperature by an average amount of 1.2 K in the daytime. This estimate is made using the offline energy balance diagnostics and Equation 6, where $\Delta a$ is citywide equivalent albedo change associated with the use of solar PV roofs. The aggregated potential of 100% solar PV roof, 100% street vegetation and 100% reflective pavement is 2.8 K under the RCP4.5 scenario.

The above discussion focuses on mitigation of the daytime temperature. Unfortunately, none of the strategies are effective in eliminating the nighttime UHI. Conversion to cool roofs reduces building storage of solar radiation in the daytime and subsequent heat release at night, contributing to nighttime cooling by an average amount of 0.7 ± 0.03 K (Figures 3d & 4d). This is not nearly enough to offset the UHI (3.0 ± 0.1 K; Figures 3b & 4b) or the GHG warming (2.6 ± 0.3 K) expected near the end of the century under the RCP4.5 scenario. The lack of nighttime cooling underscores the importance of increasing

resilience and preparedness to cope with heat stress (Stone et al., 2012; Revi et al., 2014), in addition to re-engineering the city landscape to achieve daytime temperature reduction. In CLM, AC is switched on when the interior temperature of a building is greater than 24.5$^{\circ}$C. The daytime AC energy saving in the WHT simulations is 26.0 and 24.8 W m$^{-2}$, or 55% and 43% under the RCP4.5 and RCP8.5 scenarios, respectively, in comparison to the CTR simulations, which is more than enough to support the AC energy use for cooling at night (Table 2).



The above wedge strategy is applied to the surface UHI. To find out if the linear supposition can also be extended to air UHI, we also conducted a meta-analysis based on the published studies that presented modeled air UHI results of both combined mitigation strategies and individual strategies. We calculated the sum of the cooling benefit from each simulation of

individual strategy (as is shown in the x-axis of Figure 6), and compared it with the total cooling benefit from the combined simulation (as is shown in the y-axis of Figure 6). These data collected from the literature include simulations of cool roof, green roof, and solar PV. For example, Georgescu et al. (2014) showed that if cool roof and green roof are implemented individually in California, the cooling benefit is 1.45 and 0.24 K, respectively. The linear supposition principle estimates that if both methods are put in place, the overall cooling is the sum of the individual benefit which is 1.69 K (1.45 K + 0.24 K).

For comparison, the simulated cooling for simultaneous implementation of these two mitigation methods is 1.66 K. Figure 6 demonstrates that the UHI mitigation wedge method (linear supposition) holds valid for the air UHI as well. This confirms that the wedge idea can provide a reasonable estimate of the maximum potential cooling benefit of multiple strategies collectively. Although not directly related to heat exposure, the surface UHI yields robust conclusions and implications when used as a universal metric consistently across different cities.

**4 Conclusions**

The UHI intensity is projected to increase in future warmer climates, partly due to higher anthropogenic heat release from AC energy use to cope with the GHG-induced warming. Our modeling analyses favor cool roofs as the preferred method for urban heat mitigation in comparison to green roofs, street vegetation, and reflective pavement. By adopting highly reflective roofs citywide, almost all the selected cities in the USA and in southern Canada are transformed into "white oases". Cool

roofs also bring large daytime AC energy savings in future climate scenarios. A UHI mitigation wedge strategy consisting of 50% cool roof, 50% green roof, street vegetation and reflective pavement has the potential to reduce the urban daytime surface temperature by 5.7 K in the summer from the unmitigated urban scenario. Unfortunately, none of the UHI mitigation methods are effective in eliminating the nighttime UHI.

Cities are in fact engineered landscapes. The above UHI wedge strategies amount to a re-engineering of these landscapes (Supplementary Animation Movies 1 and 2). A key distinction between urban engineering and geoengineering is the scale. Unlike planetary-scale geoengineering, urban engineering impacts a much smaller areal extent (about 2.7% of the terrestrial land; Schneider et al., 2009). Planetary albedo modification can temporarily cool the global climate, but at potentially large environmental prices (Mcnutt et al., 2015). In contrast, reengineering of the urban land should have minimal negative

consequences. Reorientation of the discussion from the global-scale albedo intervention to the small-scale temperature modification can motivate local actions because the payback is immediate and direct.




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

**Acknowledgements**

This research was supported by the Ministry of Education of China (grant PCSIRT), a Princeton STEP Post-Doctoral Fellowship (to L. Z.), and a Yale University Graduate Fellowship (to N. S.). We thank Chang Cao for making the animation

15  movies. We acknowledge high-performance computing support from Yellowstone (ark:/85065/d7wd3xhc) provided by NCAR's Computational and Information Systems Laboratory, sponsored by the U. S. National Science Foundation.

30





5  **Table 1**: The simulations run and the methods used to assess the temperature mitigation strategies. The online simulation method calculated the mitigation potential of cool roofs as the difference between the WHT and CTR simulations (WHT – CTR). The offline attribution and two end-member interpolation methods used diagnostic data from the CTR simulations.

| model simulations | Albedo | | Mitigation strategy method | | | |
| --- | --- | --- | --- | --- | --- | --- |
| | CTR | WHT | Cool roofs | Street Vegetation | Green roofs | Reflective pavement |
| current (1972-2004)<br>RCP 4.5 (2071-2100)<br>RCP 8.5 (2071-2100) | default<br>(0.18 - 0.37) | 0.88 | online simulations,<br>offline attribution | two end-member<br>interpolation | offline attribution | offline attribution |



**Table 2**: The mean anthropogenic heat flux for all the selected cities and the cities in the three climate zones in the current climate, RCP 4.5 and RCP 8.5 scenario from both the control (CTR) and the cool-roof (WHT) simulations. (Units: W m$^{-2}$)

| | Daytime | | | | | | | |
|---|---|---|---|---|---|---|---|---|
| | CTR | | | | WHT | | | |
| | **Dry** | **Continental** | **Temperate** | **All** | **Dry** | **Continental** | **Temperate** | **All** |
| Current | 8.9 | 2.3 | 56.8 | 26.8 | 1.6 | 1.9 | 18.3 | 8.7 |
| RCP 4.5 | 39.3 | 7.9 | 84.1 | 47.1 | 11.8 | 4.5 | 39.9 | 21.1 |
| RCP 8.5 | 55.1 | 14.6 | 95.2 | 57.8 | 21.8 | 7.3 | 60.4 | 32.9 |
| | Nighttime | | | | | | | |
| | CTR | | | | WHT | | | |
| | **Dry** | **Continental** | **Temperate** | **All** | **Dry** | **Continental** | **Temperate** | **All** |
| Current | 5.0 | 3.7 | 16.9 | 9.5 | 4.6 | 2.7 | 17.2 | 9.2 |
| RCP 4.5 | 14.2 | 6.7 | 33.0 | 19.5 | 12.4 | 5.3 | 27.3 | 16.2 |
| RCP 8.5 | 23.4 | 11.8 | 47.8 | 29.6 | 21.0 | 9.1 | 41.8 | 25.6 |




**List of Figure Captions**

**Figure 1**: Attribution of summer mean UHI intensity during 2071-2100 under the RCP 4.5 scenario for the control (CTR) run. a, b, c, d: daytime; e, f, g, h: nighttime. a, e: dry climate; b, f: continental climate; c, g: temperate climate; d, h: all selected cities. The radiative forcing term results mostly from albedo differences between urban and rural land in the daytime and from small differences in surface emissivity at night. Error bars are ± 1 standard error.

**Figure 2:** Scatter plot between the daytime UHI intensity computed online with the CLM and that calculated offline using the surface energy balance diagnostic data. Filled and open symbols denote results from control (CTR) and cool-roof (WHT) simulations, respectively. Red, green and blue colors denote dry (Dry), continental (Cont) and temperate (Temp) climate region, respectively. Circle, triangle and square symbols denote current climate (Cur), RCP4.5 (RCP45) and RCP8.5 scenario (RCP85), respectively.

**Figure 3**: Urban heat island intensity from the control (CTR, a, b) and cool-roof (WHT, c, d) simulations. a, c: daytime; b, d: nighttime; Red, green and blues bars denote dry, continental, and temperate climate zone, respectively. Error bars are ± 1 standard error.

**Figure 4:** Maps of summer mean urban heat island intensity during 2071-2100 under the RCP4.5 scenario. a: daytime control (CTR) simulation; b: nighttime CTR simulation; c: daytime cool-roof (WHT) simulation; d: nighttime WHT simulation. Red and blue symbols denote positive and negative UHIs, respectively.

**Figure 5:** A UHI strategy consisting of three mitigation wedges under the RCP4.5 scenario. a: cool roof, street vegetation, and reflective pavement; b: green roof, street vegetation, and reflective pavement. The horizontal line marks the mean midday rural surface temperature of all the 57 cities in the current climate conditions, and other temperatures are mean values relative to this rural background.

**Figure 6:** Comparison between combined cooling benefit (combined simulation results) and sum of the components (sum of individual simulation results). Black symbols: cool roof + solar panel roof; green symbols: cool roof + green roof; circles: Salamanca et al. (2016); squares: Taha et al. (1997); stars: Stone et al. (2014); triangles: Georgescu et al. (2014)



**Figure 1: Attribution of summer mean UHI intensity during 2071-2100 under the RCP 4.5 scenario for the control (CTR) run. a, b, c, d: daytime; e, f, g, h: nighttime. a, e: dry climate; b, f: continental climate; c, g: temperate climate; d, h: all selected cities. The radiative forcing term results mostly from albedo differences between urban and rural land in the daytime and from small differences in surface emissivity at night. Error bars are ± 1 standard error.**





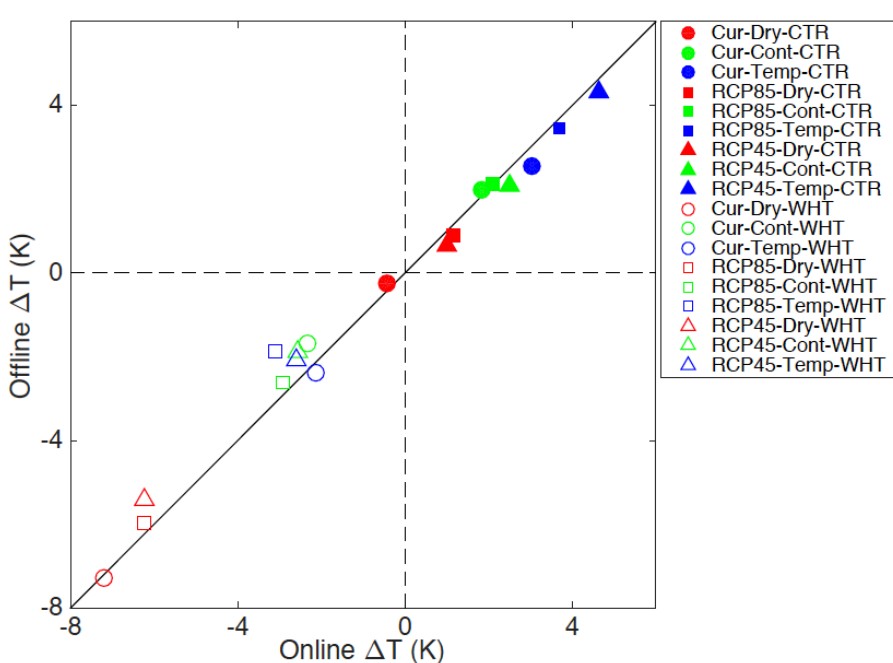

**Figure 2: Scatter plot between the daytime UHI intensity computed online with the CLM and that calculated offline using the surface energy balance diagnostic data. Filled and open symbols denote results from control (CTR) and cool-roof (WHT) simulations, respectively. Red, green and blue colors denote dry (Dry), continental (Cont) and temperate (Temp) climate region, respectively. Circle, triangle and square symbols denote current climate (Cur), RCP4.5 (RCP45) and RCP8.5 scenario (RCP85), respectively.**





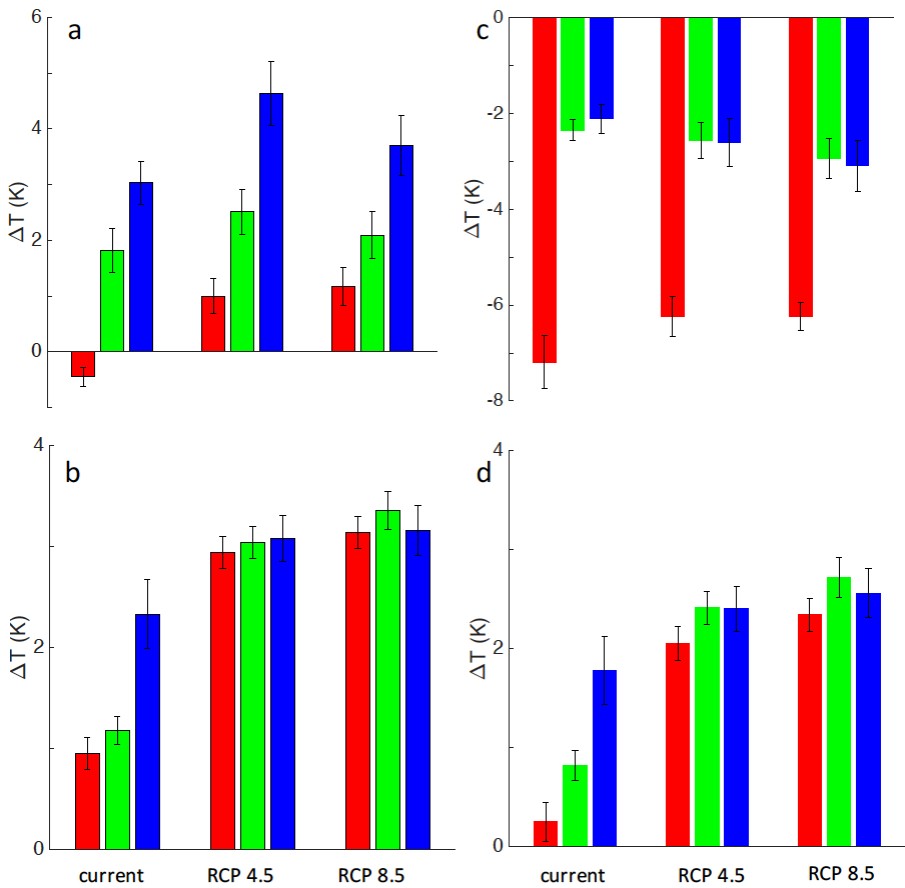

**Figure 3: Urban heat island intensity from the control (CTR, a, b) and cool-roof (WHT, c, d) simulations. a, c: daytime; b, d: nighttime; Red, green and blues bars denote dry, continental, and temperate climate zone, respectively. Error bars are ± 1 standard error.**





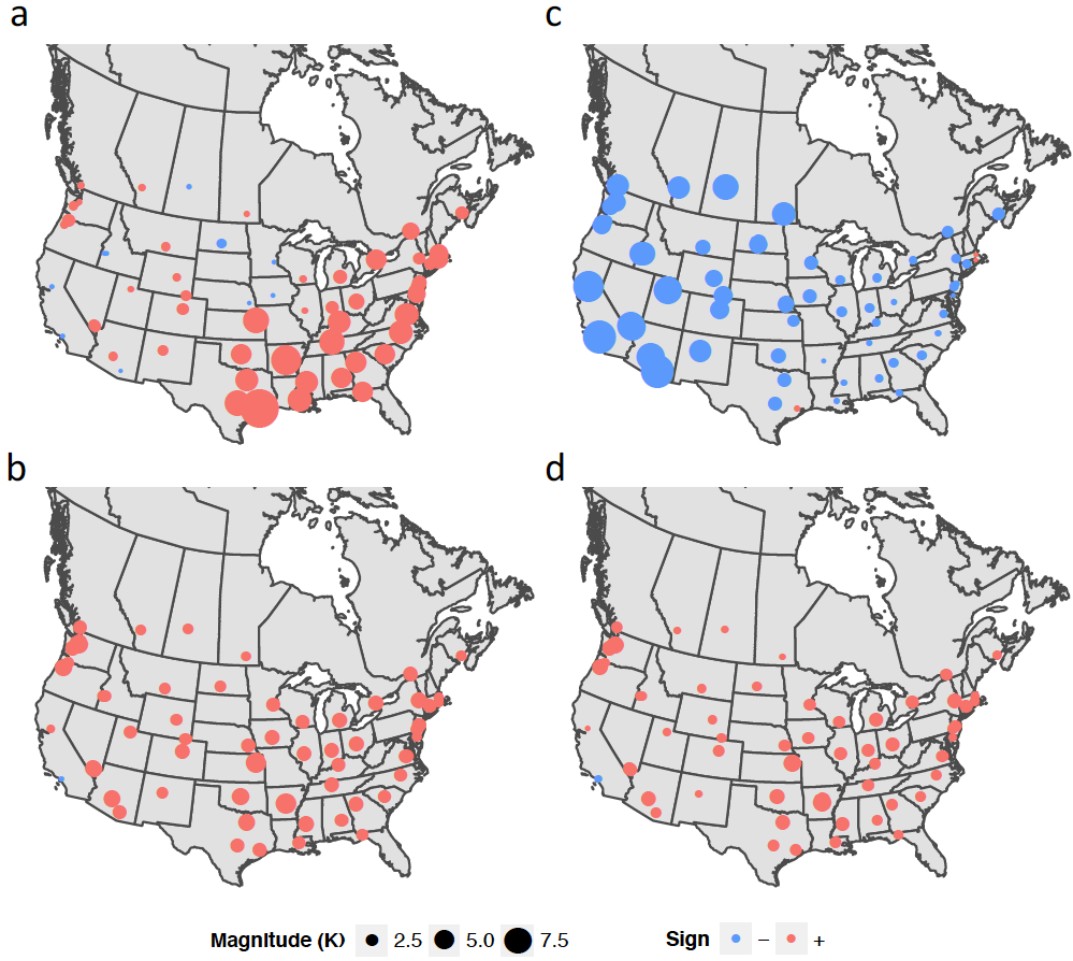

**Figure 4: Maps of summer mean urban heat island intensity during 2071-2100 under the RCP4.5 scenario. a: daytime control (CTR) simulation; b: nighttime CTR simulation; c: daytime cool-roof (WHT) simulation; d: nighttime WHT simulation. Red and blue symbols denote positive and negative UHIs, respectively.**


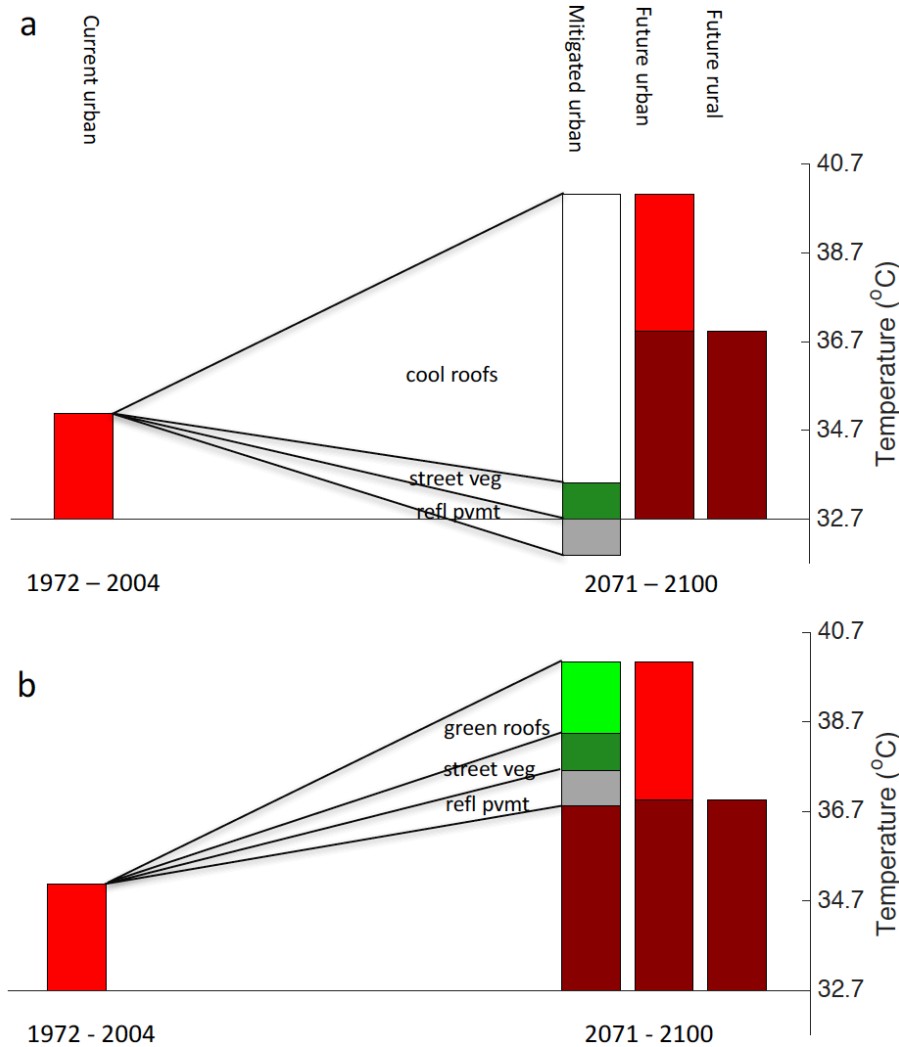

5   **Figure 5: A UHI strategy consisting of three mitigation wedges under the RCP4.5 scenario. a: cool roof, street vegetation, and reflective pavement; b: green roof, street vegetation, and reflective pavement. The horizontal line marks the mean midday rural surface temperature of all the 57 cities in the current climate conditions, and other temperatures are mean values relative to this rural background.**





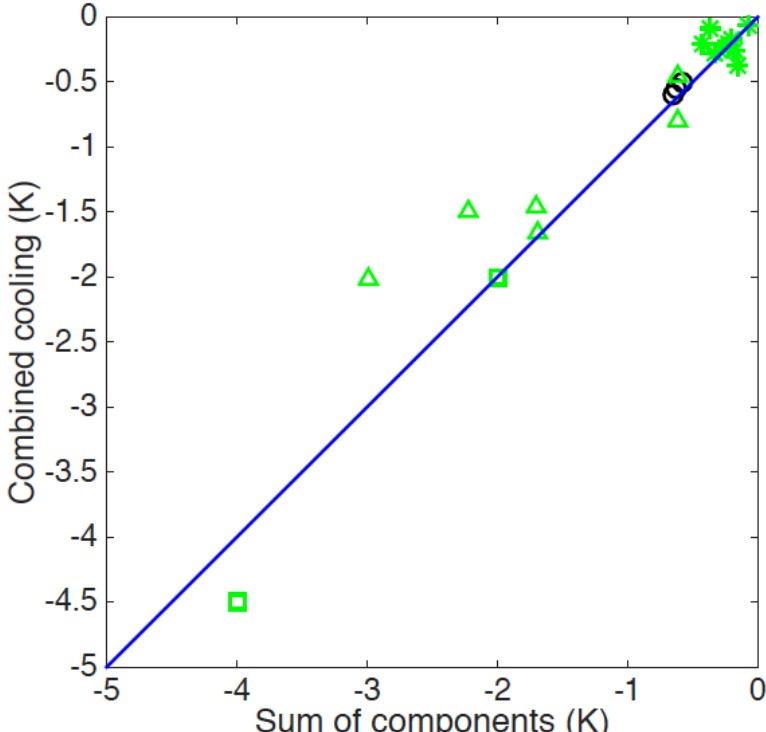

**Figure 6: Comparison between combined cooling benefit (combined simulation results) and sum of the components (sum of individual simulation results). Both are measured by screen-height air temperature. Black symbols: cool roof + solar panel roof; green symbols: cool roof + green roof; circles: Salamanca et al. (2016); squares: Taha et al. (1997); stars: Stone et al. (2014); triangles: Georgescu et al. (2014)**

