# Peer review of "A wedge strategy for mitigation of urban warming in future climate scenarios"

_Atmospheric Chemistry and Physics, 2016_

## Referee Comment (RC1) · Anonymous Referee #2 · 12 Feb 2017

Overall

In this study, Zhao et al., combined offline mathematical attribution, and online model simulations to compare the effectiveness of four urban heat island mitigation strategies, including cool roof, green roof, street vegetation, and reflective pavement. Their major finding was that cool roof (albedo $\sim$0.9) was the most effective way to reduce urban daytime temperature, although none of the strategy was able to substantially reduce nighttime temperature. Overall, this is a well-written, high-quality, scientifically significant paper. Below, I have several concerns about methodology and underlying model assumption and many specific comments that may improve the wording.

Major comments

First, this study conducted CESM offline simulations driven by climate outputs from

fully coupled CESM runs. The justification was that "urban land unit in CLM comprise only a small areal fraction of each grid cell, changes in urban temperature would lead to negligible changes in grid cell mean temperature". I think this is totally dependent on model resolution. Fine resolution CESM can run at about 25kmx30 km. In this case, big cities (like New York city) will almost equal to the size of the grid cell. The change of urban temperature will substantially affect mean air temperature for this particular grid cell, and also adjacent areas.

Secondly, model simulations covered 1972-2004 and 2071 to2100. I wondered how to restart the model for the 2071-2100 periods (restart from state variables in the end of 2004?). The state variables are not consistent between the two time segments. Maybe one can assume that the size of the city is fixed, that's probably the default configuration of CLM urban module. But, the vegetated land unit change dramatically from 2005 to 2070, because of for example $CO_2$ fertilization effect, fire, natural vegetated land conversion. In either case, we expect to see significant changes of vegetated land surface properties from 2005 to 2070.

Thirdly, for the street vegetation strategy. I agree that the two end member interpolation is a feasible idea, given that CLM does not mix sub grid land units (such as natural forest and city). But, there is an implicitly assumption for this approach. Street vegetation composition must be the same as surrounding vegetated land unit. In CLM each vegetated land unit is further divided into multiple plant functional types (PFT). Normally, it is a mix of forest and grassland. I wondered what is a typical forest/grass ratio in a real street vegetation setup? What's the difference of vegetation composition between the city and rural area. CLM has vegetation composition (17 PFT) for each grid cell. Will be great, if the author can show that they are consistent with a real street vegetation, at least for forest/grass ratio, since forest and grass are distinct in terms of albedo and roughness.

Finally, for some concepts, better to shortly define them at their first appearance, just in case casual readers might also be interested in this paper.

[Figure]

Specific comments

P1L11 compounded -> exacerbated

P1L12 high temperature problem -> heat stress problem

P1.L13 various urban heat mitigation strategies

P1L15 white oases, briefly define what is "white oases"

P1 L22 urban residents

P1.L22 urban heat island, briefly define what is "urban heat island"

P2L1 trials -> experiments

P1L3 at the scale of individual buildings. Specifically, . . .. . .

P2L9 financially challenging

P2L10 these methods on temperature reduction

P2L13 mitigation strategies

P2L15 high computational demand

P2L20 Looks like meso-scale weather forest model has been successfully applied to whole continental US, then the first sentence of this paragraph doesn't make sense.

P2L28 GCMs can provide usefully knowledge to city planners about

P3L1 on offsetting

P3L27 15 natural plant functional types

P5L18 climate scenarios (including . . .. . .)

P6L5 the current version, remove

P6L25 what are QS, QAH

P7L21 compared eq. 6 with eq. 2, looks like the reflective pavement strategy only considered direct effect of albedo change on temperature. I wondered, why reflective pavement ignored the second term in eq. 2?

P8L14. Figure 2 contains three climate zones (dry, continental, temperate), three scenarios (current, RCP4.5, RCP8.5), two setup (default and cool roof strategy). 3x3x2=18 combinations. What about street vegetation strategy?

Figure 5. Street vegetation and reflective pavement are they compatible? I guess, they are partially exclusive? High fraction of street vegetation will partly overlap with the pavement?

---

## Referee Comment (RC2) · Anonymous Referee #3 · 22 Apr 2017

General comments

In this article, the authors examine by means of online and offline climate simulations the effectiveness of four urban heat island mitigation strategies, namely, cool roofs, street vegetation, green roofs, and reflective pavement for three climate regions in the US and southern Canada under present and future warmer climates. Their main result is that by adopting highly reflective roofs, almost all the cities in the US and southern Canada are transformed into daytime "white oasis" (i.e., the city surface is surrounded by a hotter landscape) for the summer months of 2071-2100. Although all the mitigation strategies decrease daytime temperature, none of them was able to significantly reduce nighttime temperature.

Major concerns First, the impact of cool roofs depends of the roof areal fraction that is

in average 48.8% for the 57 selected cities. I think this value is typical for European cities but not for US cities, which are much less compact. How the roof areal fraction was estimated?

Second, The impact of reflective pavement was calculated based on the same offline method used for the green roofs. I know that it is not possible to calculate directly the cooling benefit of green roofs because CLM does not have a green roof model implemented. However, the impact of reflective pavements can be easily simulated (online) by incrementing the road albedo. I wonder why the authors did not estimate the cooling benefits of reflective pavements by means of direct simulations.

Third, model outputs at 13:00 and 01:00 LT are used to represent daytime and night-time conditions, respectively. I wonder why the authors did not include more hours in the analysis of the results.

---

## Author Response (AR1)

**Response to RC1 on *Atmospheric Chemistry and Physics* Manuscript acp-2016-1046
"A wedge strategy for mitigation of urban warming in future climate scenarios"**

**Review #2**

1. "**Overall**
In this study, Zhao et al., combined offline mathematical attribution, and online model simulations to compare the effectiveness of four urban heat island mitigation strategies, including cool roof, green roof, street vegetation, and reflective pavement. Their major finding was that cool roof (albedo ~ 0.9) was the most effective way to reduce urban daytime temperature, although none of the strategy was able to substantially reduce nighttime temperature. Overall, this is a well-written, high-quality, scientifically significant paper."

Thank you.

2. "**Major comments**
First, this study conducted CESM offline simulations driven by climate outputs from fully coupled CESM runs. The justification was that "urban land unit in CLM comprise only a small areal fraction of each grid cell, changes in urban temperature would lead to negligible changes in grid cell mean temperature". I think this is totally dependent on model resolution. Fine resolution CESM can run at about 25kmx30 km. In this case, big cities (like New York city) will almost equal to the size of the grid cell. The change of urban temperature will substantially affect mean air temperature for this particular grid cell, and also adjacent areas."

We thank the reviewer for the question. All the CESM offline simulations were conducted at the resolution of $0.9^o$ latitude × $1.25^o$ longitude. The selected cities in this study comprise less than 20% of the grid cell. Therefore, the changes in the temperature of those cities would affect minimally the mean air temperature of the corresponding grid cells. In response to the reviewer's comment, we now have added the following text: "All the simulations were run at a horizontal resolution of $0.9^o$ × $1.25^o$ (latitude × longitude)." (Page 4 Line 11-12)
"These cities comprise less than 20% of the grid cell area." (Page 4 Line 24)

3. "Secondly, model simulations covered 1972-2004 and 2071 to2100. I wondered how to restart the model for the 2071-2100 periods (restart from state variables in the end of 2004?). The state variables are not consistent between the two time segments. Maybe one can assume that the size of the city is fixed, that's probably the default configuration of CLM urban module. But, the vegetated land unit change dramatically from 2005 to 2070, because of for example $CO_2$ fertilization effect, fire, natural vegetated land conversion. In either case, we expect to see significant changes of vegetated land surface properties from 2005 to 2070."

We agree with the reviewer that the vegetated land unit changes dynamically in response to future scenarios because of $CO_2$ fertilization effect, fire, vegetation conversion, etc. In our simulations, dynamic land surface input data corresponding to each RCP scenario were used. The reviewer is also correct that the urban sizes were kept fixed (default configuration of CLM urban module) in our simulations.

For the 2071 – 2100 periods, the model was not restarted from 2071. Actually, the model simulations for future climate scenarios were run from 2005 to 2100, after 600 years of spin-up. For the purpose of comparing current climate and future warmer climates, we only selected the data of the last 30 years in this century to analyze. In the original manuscript, we pointed out:
"For the two future scenarios, the model was forced by atmospheric outputs from fully-coupled runs of the CESM (years 2005 to 2100)." (Page 4 Line 4-5)

In order to improve clarity, we have added following text:

"For the current climate, the model was run for 33 years after a 60-year spin-up, driven by a reconstructed climatology from 1972 to 2004 (Qian et al., 2006). For the two future scenarios, the model was forced by atmospheric outputs from fully-coupled runs of the CESM and run for 96 years from 2005 to 2100 after a 600-year spin-up. Dynamic land surface input data corresponding to each future scenario are used in the simulations. It should be noted here that although the vegetated land unit changes dynamically to the climate scenarios, the urban land is kept fixed in the default configuration of the current version of CESM." (Page 4 Line 3-8)

4. "Thirdly, for the street vegetation strategy. I agree that the two end member interpolation is a feasible idea, given that CLM does not mix sub grid land units (such as natural forest and city). But, there is an implicitly assumption for this approach. Street vegetation composition must be the same as surrounding vegetated land unit. In CLM each vegetated land unit is further divided into multiple plant functional types (PFT). Normally, it is a mix of forest and grassland. I wondered what is a typical forest/grass ratio in a real street vegetation setup? What's the difference of vegetation composition between the city and rural area. CLM has vegetation composition (17 PFT) for each grid cell. Will be great, if the author can show that they are consistent with a real street vegetation, at least for forest/grass ratio, since forest and grass are distinct in terms of albedo and roughness."

We thank the reviewer for the suggestion. Our simple two end-member interpolation method to calculate the surface temperature change associated with street vegetation does have an implicit assumption that the street vegetation composition is same as its surrounding rural land. We have noted in the original manuscript that "the calculation assumes that street vegetation consists of native plant species having the same species compositions in the adjacent rural land" (Page 6 Line 22-24). It is correct that in CLM vegetated land unit of each grid cell consists of up to 17 plant functional types (PFT) including both trees and grass. In CLM, the average urban tree/grass ratio for the selected cities in this study is 1.9 with different average ratios in different climate zones

(temperate: 1.8; continental: 2.5; dry: 1.2), which is in line with the real urban vegetation composition estimated using remote sensing techniques (Myeong et al., 2001; Nowak and Greenfield, 2012).

We agree with the reviewer that the street vegetation composition in the city could be different from that in the rural area. Therefore, the street vegetation composition is a potential source of uncertainty to our two end-member interpolation method of evaluating the street vegetation strategy. In response to the reviewer's concern and to caution the readers, we have modified the text in the manuscript as below:

"In CLM, the average tree-to-grass ratios in the surrounding rural land of selected cities are 1.8, 2.5 and 1.2 for the temperate, continental and dry climate zones, respectively. These numbers are in line with the real urban forest-to-grass ratio estimated using remote sensing techniques (Myeong et al., 2001; Nowak and Greenfield, 2012)." (Page 6 Line 24-26) and,

"Another potential source of uncertainty to this method is the street vegetation composition. Cities that have different vegetation composition from their surrounding rural landscapes may generate slightly different cooling than predicted here." (Page 11 Line 15-17)

5. "Finally, for some concepts, better to shortly define them at their first appearance, just in case casual readers might also be interested in this paper."

Thanks for the good suggestion. We have shortly defined certain concepts at their first appearance throughout the manuscript.

Specific Comments

6. "P1L11 compounded -> exacerbated"
Done.

7. "P1L12 high temperature problem -> heat stress problem"
Done.

8. "P1.L13 various urban heat mitigation strategies"
Done.

9. "P1L15 white oases, briefly define what is "white oases""
Done. The text has been modified to: "almost all the cities in the United States and southern Canada are transformed into "white oases" – cold islands caused by cool roofs at midday" (Page 1 Line 15)

10. "P1 L22 urban residents"
Done.

11. "P1.L22 urban heat island, briefly define what is "urban heat island""
Done. The text has been edited as: "These risks are further amplified for urban residents because of the urban heat island effect, a phenomenon in which surface temperatures are higher in urban areas than in surrounding rural areas (Grimmond, 2007)" (Page 1 Line 22-23)

12. "P2L1 trials -> experiments"
Done.

13. "P1L3 at the scale of individual buildings. Specifically, ……"
Done.

14. "P2L9 financially challenging"
Done.

15. "P2L10 these methods on temperature reduction"
Done.

16. "P2L13 mitigation strategies"
Done.

17. "P2L15 high computational demand"
Done.

18. "P2L20 Looks like meso-scale weather forest model has been successfully applied to whole continental US, then the first sentence of this paragraph doesn't make sense."
The study that successfully applied to the continental US (Georgescu et al. 2014) is an extreme and rare case of using meso-scale weather forecast model. In addition, the temporal scale of its simulations is short-term because of high computational demand.

19. "P2L28 GCMs can provide usefully knowledge to city planners about"
Done.

20. "P3L1 on offsetting"
Done.

21. "P3L27 15 natural plant functional types"
Done.

22. "P5L18 climate scenarios (including ……)"
Done.

23. "P6L5 the current version, remove"
Done.

24. "P6L25 what are QS, QAH"

Thanks. $Q_s$ is the stored heat in the canopy, and $Q_{AH}$ is the urban anthropogenic heat release. Now the notations of $Q_s$ and $Q_{AH}$ have been added in the text.

25. "P7L21 compared eq. 6 with eq. 2, looks like the reflective pavement strategy only considered direct effect of albedo change on temperature. I wondered, why reflective pavement ignored the second term in eq. 2?"

The second term in Eq. 2 represents the contribution of evapotranspiration to temperature reduction. We consider that reflective pavement would only affect the surface albedo, and would not change the surface evapotranspiration (associated with Bowen ratio), because the reflective pavement strategy in this study is restricted to only impervious surfaces in the city.

26. "P8L14. Figure 2 contains three climate zones (dry, continental, temperate), three scenarios (current, RCP4.5, RCP8.5), two setup (default and cool roof strategy). 3x3x2=18 combinations. What about street vegetation strategy?"

Figure 2 shows the comparison of online and offline estimates. Because CLM does not explicitly represents street vegetation in the urban land unit, we cannot use the online approach (direct modeling) to estimate the cooling effect of street vegetation. Therefore, both street vegetation and green roof effects are not shown in Figure 2.

27. "Figure 5. Street vegetation and reflective pavement are they compatible? I guess, they are partially exclusive? High fraction of street vegetation will partly overlap with the pavement?"

In this study, we restrict the reflective pavement strategy to impervious surfaces only and the street vegetation to pervious surfaces only in the city. When we estimate their cooling effects, we use the areal fraction of impervious and pervious surface of each city in the model for reflective pavement and street vegetation respectively. Therefore, the street vegetation and reflective pavement strategy in this study are not mutually exclusive. In order to avoid the confusion, we now have added the following text:
"Only impervious surface in each urban land unit is considered to convert into reflective pavement.". (Page 8 Line 2-3)

**Review #3**

1. "Major concerns First, the impact of cool roofs depends of the roof areal fraction that is in average 48.8% for the 57 selected cities. I think this value is typical for European cities but not for US cities, which are much less compact. How the roof areal fraction was estimated?"

For each of the 57 selected cities, we used the default number of roof areal fraction in the urban land unit of the grid cell prescribed in the CLM land surface data. In CLM, all morphological properties of roof/wall/road such as roof areal fraction, height-to-width ratio, average building height, and pervious floor fraction are provided by Jackson et al. (2010). These properties are basically remote sensing and GIS based estimates for global climate modeling. Details of the methodology used to estimate these urban characteristics can be found in Jackson et al. (2010). In this study, we assumed that the penetration rate of both cool and green roof is 100% for all selected cities. Therefore 48.8% is the average number of the prescribed roof areal fractions in the CLM for all the selected cities. In response to the reviewer's question and to clarify for the readers, we have added the text in the manuscript as below:

"The thermal (such as heat capacity and thermal conductivity), radiative (such as albedo and emissivity) and morphological (such as height-to-width ratio, roof areal fraction, average building height and pervious ground fraction) characteristics of these canyon components are provided by Jackson et al. (2010) for each grid cell." (Page 3 Line 26-29) and,

"The morphological properties were kept unchanged in the CTR and WHT simulations. Of the 57 selected cities, the average prescribed roof areal fraction in CLM 4.0 is 48.8%" (Page 6 Line 5-6)

2. "Second, the impact of reflective pavement was calculated based on the same offline method used for the green roofs. I know that it is not possible to calculate directly the cooling benefit of green roofs because CLM does not have a green roof model implemented. However, the impact of reflective pavements can be easily simulated (online) by incrementing the road albedo. I wonder why the authors did not estimate the cooling benefits of reflective pavements by means of direct simulations."

We agree with the reviewer that the cooling benefits of reflective pavements can be simulated online as cool roofs by incrementing the impervious road albedo. We did a comparison between the online and offline estimated $\Delta T$ for cool roofs (Figure 2), and their excellent agreement strongly supports the accuracy of our offline attribution method. Considering that reflective pavements is nearly identical to cool roofs with respect to perturbation of surface energy balance expect the amount of albedo elevated, we believe that the online simulated cooling benefits of reflective pavements would be very close to our offline estimates. In addition, the offline way is much more computationally inexpensive compared to the online way because there will be no need to

set up another group of simulations for reflective pavements in present climate and future RCP scenarios. This is also in line with one of our objective statements of this study: "We propose a new method to assess the mitigation strategies, which is based on a theoretical understanding of the surface energy balance and is unconstrained by computational demand" (Page 3 Line 7-8).

In response to the reviewer's concern, we conducted another simulation to directly simulate the cooling benefits of reflective pavements under current climate. Results show an excellent agreement between the online and offline $\Delta T$ of reflective pavements. The $R^2$ value between the two methods of $\Delta T$ estimations is 0.98 and the mean difference is 0.3 K. We have also noted these information in the manuscript as below:

"In order to further confirm the validity of the offline method, we conducted another side simulation to directly simulate the cooling benefits of reflective pavements under current climate. In this simulation, the albedo of impervious surface in each urban land unit of the 57 cities were raised to 0.25." (Page 8 Line 5-7) and,

"Similarly, the online simulated and offline estimated $\Delta T$ for reflective pavements also show an excellent agreement with the R2 value of 0.98 and the mean difference of 0.3 K." (Page 9 Line 2-3)

3. "Third, model outputs at 13:00 and 01:00 LT are used to represent daytime and nighttime conditions, respectively. I wonder why the authors did not include more hours in the analysis of the results."

We thank the reviewer for the question. Because our offline attribution method based on the surface energy balance is valid on the hourly scale, we used hourly data in the analysis. Model outputs at 13:00 and 01:00 local time were chosen due to two reasons. First, these two hours are close to the times of daily maximum and minimum temperature, thus giving a better representation of the diurnal range of $\Delta T$. Local time 13:00 is also near the time of solar radiation peak, which could give us the maximum potential of albedo effectiveness. Second, the performance of both CLM and our offline method in estimating $\Delta T$ was validated against MODIS observations in a previous study (Zhao et al. 2014). These two times (13:00 and 01:00 local time) were close to the MODIS overpass times.

In response to the reviewer's question, we have now added the text in the manuscript as below:

[revised manuscript text omitted]